# The Influence of Cross-Reactive T Cells in COVID-19

**DOI:** 10.3390/biomedicines12030564

**Published:** 2024-03-02

**Authors:** Peter J. Eggenhuizen, Joshua D. Ooi

**Affiliations:** Centre for Inflammatory Diseases, Department of Medicine, School of Clinical Sciences, Monash University, Clayton, VIC 3800, Australia

**Keywords:** COVID-19, SARS-CoV-2, Cross-reactive immunity, T cell, heterologous immunity

## Abstract

Memory T cells form from the adaptive immune response to historic infections or vaccinations. Some memory T cells have the potential to recognise unrelated pathogens like severe acute respiratory syndrome coronavirus 2 (SARS-CoV-2) and generate cross-reactive immune responses. Notably, such T cell cross-reactivity has been observed between SARS-CoV-2 and other human coronaviruses. T cell cross-reactivity has also been observed between SARS-CoV-2 variants from unrelated microbes and unrelated vaccinations against influenza A, tuberculosis and measles, mumps and rubella. Extensive research and debate is underway to understand the mechanism and role of T cell cross-reactivity and how it relates to Coronavirus disease 2019 (COVID-19) outcomes. Here, we review the evidence for the ability of pre-existing memory T cells to cross-react with SARS-CoV-2. We discuss the latest findings on the impact of T cell cross-reactivity and the extent to which it can cross-protect from COVID-19.

## 1. Introduction: SARS-CoV-2 and COVID-19

The *Betacoronavirus* Severe Acute Respiratory Syndrome Coronavirus-2 (SARS-CoV-2) causes coronavirus disease 2019 (COVID-19) and is responsible for the recent human pandemic [1,2]. Upon infection, the human immune system mounts an orchestrated response to contain the viral load, which initiates with the innate immune system by producing type I interferons [3,4] that are crucial in mounting a functional and effective initial response [5] and sets the premise for a successful adaptive immune response and a favourable clinical outcome [6].

## 2. Adaptive Immune Response to SARS-CoV-2

T cells form part of the adaptive immune response, and they are crucial to combat SARS-CoV-2 infection since convalescent individuals exhibit SARS-CoV-2-specific T cell memory [7]. The early involvement of CD8+ cytotoxic T cells around 7–14 days after symptom onset are critical for effectively clearing the virus, resulting in mild symptoms [8,9], and these follow a similar timeline to the humoral response [10]. Immune dysregulation during SARS-CoV-2 infection leads to poorer prognosis. Ineffective interferon signalling during acute infection and T cell dysfunction, T cell number imbalance and CD8+ lymphopenia result in more severe COVID-19 clinical outcomes [11,12]. The cellular immune response, inclusive of antigen-specific and bystander effects, is also critical for driving disease outcomes, where a type 1 CD4+ T cell phenotype is associated with viral control, less-severe disease and clearance, whereas a type 2 CD4+ T cell phenotype is associated with more severe disease outcomes [8,13,14].

The adaptive immune response to SARS-CoV-2 is antigen-specific in nature through the processing and presentation of SARS-CoV-2 epitopes bound to major histocompatibility complex (MHC) on antigen presenting cells; CD8+ T cells through their T cell receptor (TCR) recognise SARS-CoV-2 antigens presented by MHC class I and CD4+ T cells through their TCR-recognising SARS-CoV-2 antigens presented by the MHC class II. The SARS-CoV-2 epitopes responsible for driving the adaptive immune response have been studied in great detail since the SARS-CoV-2 sequence was released [12,15,16]. Antigen-specific responses have been identified across all SARS-CoV-2 proteins by both CD4+ and CD8+ T cells, and over two thousand epitopes have been identified to date [17,18,19]. The immunodominant regions of SARS-CoV-2 responsible for the majority of immune responses have been extensively studied, including those commonly shared between different HLA-typed donors [15,20]. Responses to spike antigens are both CD4+ and CD8+ T cell-dominated, with the SARS-CoV-2-specific CD4+ T cells and T follicular helper cells assisting in the production of antibodies [15,16,17,21,22,23]. Notable non-spike CD8+ T cell responses protecting from severe COVID-19 include SARS-CoV-2 nucleocapsid protein [24,25]. Other non-spike epitopes recognised by T cells are derived from the membrane protein and non-structural proteins (NSPs) [15,16,26,27,28]. Overall, individuals typically show expanded epitope-specific responses to between 17 and 19 different SARS-CoV-2 epitopes, forming approximately 0.5% of the total CD4+ T cell repertoire and 0.2% of the total CD8+ T cell repertoire [15,27]. After infection, SARS-CoV-2-specific T cells become memory T cells, which are predominately CD4+ and exhibit a central memory phenotype and T effector memory re-expressing CD45RA (TEMRA) cells [27,29,30,31]. To date, this memory pool is robust, with a half-life of approximately 200 days pointing to a slow decrease in frequency over time [27,32].

Although cellular immunity to SARS-CoV-2 predominantly arises from SARS-CoV-2-specific T cells via SARS-CoV-2 infection or vaccination, there is a growing appreciation of the contribution of antigen-specific T cell responses arising from pre-existing memory T cells from infections or vaccinations other than SARS-CoV-2 [16,33,34,35,36]. Such T cell cross-reactivity can arise through T cell receptor (TCR)-dependent mechanisms [37,38,39].

## 3. TCR-Dependent Cross-Reactivity

TCR-dependent cross-reactivity arises through T cell cross-reactivity between unrelated pathogens via TCRs that can recognise both pathogens. Initially, an infection or immunisation produces memory T cells that, upon exposure to a second, different infection, cross-react and activate the memory T cells to become effector T cells (Figure 1) [40]. This happens via a TCR on the memory T cell that can sufficiently bind to MHC, presenting either the epitope from the first pathogen or the similar epitope from the second pathogen. The mechanisms behind TCR-dependent T cell cross-reactivity are actively being explored in COVID-19, as well as any correlate of protection they may have in improving COVID-19 outcomes. This review will cover three aspects of T cell cross-reactivity to SARS-CoV-2: (1) T cell cross-reactivity and cross-protection between SARS-CoV-2 and other human coronaviruses. (2) The cross-reactive T cell response to novel SARS-CoV-2 variants in the context of pre-existing SARS-CoV-2 T cell immunity from SARS-CoV-2 vaccination or prior infection. (3) Cross-reactive T cell responses from other pathogens and vaccines, such as the influenza, measles, mumps and rubella (MMR) and Bacillus Calmette–Guérin (BCG) vaccines.

Cross-reactive T cells first arise when naïve T cells recognise, through their T cell receptor (TCR), an antigen from the vaccine or pathogen presented by the major histocompatibility complex (MHC) on antigen-presenting cells (APC) such as dendritic cells (DC). These T cells become T effectors before contracting into a T memory (T mem) phenotype. Given sufficient structural or sequence similarity between the first antigen and SARS-CoV-2, the TCR on the memory T cell can recognise the SARS-CoV-2 antigen presented by DC/APC. This activates the T cell to become a T effector and produce cross-reactive T cell responses.

## 4. T cell Cross-Reactivity between SARS-CoV-2 and Other Human Coronaviruses

Cross-reactive T cells between SARS-CoV-2 and other human coronaviruses (HCoVs) were identified early on in the pandemic in individuals unexposed to SARS-CoV-2 [16,33,34,41,42,43,44]. The less serious, seasonal HCoVs are the *Betacoronavirus* OC43 and HKU1 and *Alphacoronavirus* NL63 and 229E. Approximately 90% of the adult human population has been exposed to each of these viruses, and the four seasonal coronaviruses are responsible for 15–30% of all respiratory tract infections each year, meaning there is a great deal of potential for the pool of memory T cells to cross-react with SARS-CoV-2 [45,46]. Other more serious but less common HCoVs are Middle East respiratory syndrome coronavirus (MERS-CoV) and SARS-CoV-1. These six HCoVs share a degree of amino acid sequence homology with SARS-CoV-2 and, thus, contribute to T cell cross-reactive responses.

The seasonal HCoVs, although prevalent, do not sustain antibodies long-term, and T cell memory responses are present but generally of low magnitude, meaning humans can typically become reinfected within 12 months [47,48]. For SARS-CoV-1, responsible for the 2002–2004 SARS outbreak, memory T cell responses were detectable as long as 17 years after infection, much longer than humoral responses [33]. For MERS-CoV, a similar persistence of T cell responses over humoral responses was observed [49,50,51]. Overall, this highlights the importance of T cell memory and its potential for cross-reactivity among shared epitopes in controlling genetics-related HcoV infections, such as SARS-CoV-2.

SARS-CoV-2-specific T cells have been identified in unexposed individuals, and they are suspected to arise from memory T cell cross-reactivity from previous HcoV infections, which share key T cell epitopes [16,33,34,41,42,43,44,52,53]. A list of SARS-CoV-2 T cell epitopes shown to cross-react with other human coronaviruses is found in Table 1. Cross-reactive T cell responses have been shown to generate functional T cell responses in most but not all reports [12,33,52,54]. However, there remains debate about whether the functionality of these cross-reactive T cells can contribute to the cross-protective effect and impact clinical outcomes.

There is evidence to suggest cross-reactive T cell immunity may not always correlate with positive clinical outcomes. It has been shown that cross-reactive T cells have a low avidity for SARS-CoV-2 homologues, and low avidity T cell responses are correlated with severe COVID-19 [55,56]. This suggests that TCR engagement with peptide-MHC may not be sufficient to properly activate the cross-reactive memory T cells and turn them into robust T effectors against SARS-CoV-2. Also, there is a risk that the cross-reactive T cell repertoire may actually hinder clinical outcomes by engaging only mildly effective effectors against the infection and occupy the immunological space at the expense of more effective, higher affinity/avidity TCR clonotypes [55].

Among adults, cross-reactive T cells against HcoVs are of a low magnitude, and their persistence is not fully understood [48]. Interestingly though, among young adults and children, cross-reactive T cells and antibodies are present, particularly against the spike 2 domain, a region that is relatively conserved between HcoVs [57,58]. Conversely, among the elderly, HcoV-specific T cells and antibodies are mostly non-existent [48]. This may be a contributing factor for why COVID-19 is relatively mild in children and more severe in the elderly.

Despite evidence showing that cross-reactive T cells are less effective in combatting SARS-CoV-2 infection, there is evidence to suggest that cross-reactive T cells can protect from severe COVID-19. In the context of previous recent HcoV infections, the HcoV-specific T cells are able to cross-react and protect against subsequent infection with SARS-CoV-2, which leads to less severe COVID-19 [59]. There may be a time-dependent effect for cross-protection by recent HcoV infection given that seasonal HcoV memory T cells are relatively short-lived. Another study associated protection from COVID-19 with cross-reactive T cells as higher frequencies of cross-reactive memory T cells against SARS-CoV-2 nucleocapsid were present in patients who remained PCR-negative despite exposure to SARS-CoV-2 compared to PCR-positive SARS-CoV-2-exposed individuals [60]. Thus, there is potential for cross-reactive T cells to result in asymptomatic COVID-19.

Another major contributor to HcoV cross-reactivity with SARS-CoV-2 arises from epitopes within the NSPs. Given that the NSPs are relatively well conserved between HcoVs and by harnessing the potential of cross-reactive T cell immunity, the shared homology between NSPs can be utilised for the development of a pan-coronavirus vaccine that has the potential to protect from seasonal HcoVs, SARS-CoV-2 and any future coronaviruses that may arise [61]. There has been much effort to define the cross-reactive epitopes and their associated TCRs that can recognise a broad range of HcoVs and even other zoonotic coronaviruses, which pose a risk to humans [25,56,62,63,64,65,66,67,68,69,70]. Pan-coronavirus vaccines are important for minimising the risk of further pandemics caused by coronaviruses. By utilizing cross-reactive T cell responses driven by non-spike epitopes such as NSPs, such an approach can protect from a variety of HcoVs as well as SARS-CoV-2.

The SARS-CoV-2 spike and nucleocapsid proteins are responsible for a major part of the natural adaptive immune response to SARS-CoV-2, with the spike notably being the antigen used in SARS-CoV-2 vaccines. T cell cross-reactivity to the SARS-CoV-2 spike and nucleocapsid proteins has been implicated in cross-protective immunity. The spike and nucleocapsid epitopes of SARS-CoV-2 share significant homology with other HcoVs. In a humanised mouse model, prior infection with the HcoV OC43 protected mice against disease when infected with SARS-CoV-2. Cross-protection occurred due to CD4+ and CD8+ T cell cross-reactivity to key spike and nucleocapsid epitopes [71]. In humans, a common HLA type, HLA-B*15:01, has been shown to bind SARS-CoV-2 and multiple HcoV epitopes and produce cross-reactive memory T cell responses [72]. This immunodominant, the cross-reactive epitope is likely the reason for the strong association between individuals with HLA-B*15:01 and asymptomatic SARS-CoV-2 infection [73].

There are reports that SARS-CoV-1 and MERS-CoV memory T cells can cross-react with SARS-CoV-2, which is likely due to their close phylogenetic association and high sequence homology [33,74,75]. Both SARS-CoV-1 and MERS-CoV infections result in short-lived B cell and antibody responses but encouragingly long-lasting T cell memory responses up to 18 years post-infection [33,76,77]. However, upon closer inspection, there was low homology between the immunodominant SARS-CoV-2 epitopes and their homologues in SARS-CoV-1 [33,42,78,79]. This may mean that despite the high degree of homology between SARS-CoV-1, MERS-CoV and SARS-CoV-2, as well as the detectable and durable cross-reactive T cell responses already identified in multiple studies, the particular cross-reactive epitopes resulting in an effective immune response against SARS-CoV-2 are not covered by such cross-reactivity. As such, a cross-protective effect arising from such cross-reactivity may be insufficient, although the extent of any cross-protective effect in COVID-19 outcomes requires further research. Given that SARS-CoV-1 was a relatively isolated, historic outbreak from 2002 to 2004, the biological importance holds less relevance in terms of the current public health landscape.

**Table 1 biomedicines-12-00564-t001:** SARS-CoV-2 T cell epitopes known to cross-react with human coronavirus epitopes. This list is not exhaustive.

HLA Association	SARS-CoV-2 Epitope	SARS-CoV-2 Sequence	Reference
Cross-reactive Spike Epitopes			
HLA-DP	S_355–364_	RKRISNCVAD	[63]
HLA-DR	S_506–525_	QPYRVVVLSFELLHAPATVC	[63]
NA	S_556–564_	NKKFLPFQQ	[80]
NA	S_770–777_	IAVEQDKN	[80]
NA	S_810–816_	KPSKRS	[57]
NA	S_817–824_	FIEDLLFN	[80]
HLA-DP	S_816–830_	SFIEDLLFNKVTLAD	[42,44,56,63]
NA	S_851–856_	CAQKFN	[57]
NA	S_901–906_	QMAYRF	[57]
HLA-B*15:01	S_919–927_	NQKLIANQF	[73]
HLA-A*02:01	S_976–984_	VLNDILSRL	[81]
NA	S_997–1002_	ITGRLQ	[57]
HLA-DP	S_981–1000_	LSRLDKVEAEVQIDRLITGR	[63]
NA	S_1040–1044_	VDFCG	[57]
NA	S_1148–1157_	FKEELDKYFK	[80,82]
NA	S_1150–1156_	EELDKYF	[80,82]
NA	S_1205–1212_	KYEQYIKW	[57]
NA	S_1206–1220_	YEQYIKWPWYIWLGF	[42]
HLA-A*24	S_1207–1215_	QYIKWPWYI	[43]
Cross-reactive NSP Epitopes			
HLA-A*02:01	NSP1 (ORF1_84–92_)	VMVELVAEL	[81]
HLA-A*01:01	NSP3 (ORF1_1637–1646_)	TTDPSFLGRY	[43]
HLA-A*02:01	NSP5 (ORF1_3467–3475_)	VLAWLYAAV	[81]
HLA-B*08	NSP5 (ORF1_3361–3369_)	TPKYKFVRI	[43]
HLA-A*02:01	NSP6 (ORF1_3690–3698_)	KLKDCVMYA	[83]
HLA-B*35	NSP7_36–50_	HNDILLAKDTTEAFE	[33]
NA	NSP7_26–40_	SKLWAQCVQLHNDIL	[33]
HLA-A*02:01	NSP8 (ORF1_4032–4040_)	MLFTMLRKL	[81]
NA	NSP8 (ORF1_3976–3990_)	VLKKLKKSLNVAKSE	[42]
HLA-B*08	NSP10 (ORF1_4344–4352_)	DLKGKYVQI	[43]
NA	NSP12 (ORF1_5246–5260_)	LMIERFVSLAIDAYP	[42]
HLA-A*24	NSP12 (ORF1_5137–5145_)	FYAYLRKHF	[83]
NA	NSP12 (ORF1_5136–5150_)	EFYAYLRKHFSMMIL	[42]
NA	NSP12 (ORF1_4966–4980_)	KLLKSIAATRGATVV	[42]
HLA-A*02:01	NSP12 (ORF1_4515–4523_)	TMADLVYAL	[81]
NA	NSP13 (ORF1_5881–5895_)	NVNRFNVAITRAKVG	[42]
HLA-A*03	NSP13 (ORF1_5455–5463_)	KLFAAETLK	[43]
NA	NSP13 (ORF1_5361–5375_)	TSHKLVLSVNPYVCN	[42]
HLA-B*40	NSP14 (ORF1_6219–6228_)	IEYPIIGDEL	[43]
NA	NSP15 (ORF1_6751–6765_)	LDDFVEIIKSQDLSV	[43]
NA	ORF8_43–57_	SKWYIRVGARKSAPL	[43]
NA	ORF7a_90–104_	QEEVQELYSPIFLIV	[43]
HLA-B*40	ORF7a_40–49_	YEGNSPFHPL	[43]
HLA-DR	ORF6_26–40_	IWNLDYIINLIIKNL	[43]
HLA-A*01	ORF6_20–31_	RTFKVSIWNLDY	[43]
Cross-reactive Nucleocapsid Epitopes			
HLA-DR	N_50–64_	ASWFTALTQHGKEDL	[43]
HLA-B*07	N_101–120_	MKDLSPRWYFYYLGTGPEAG	[33,43]
HLA-B*07:01	N_105–113_	SPRWYFYYL	[25,26,65,69,83]
HLA-DR	N_127–141_	KDGIIWVATEGALNT	[43]
NA	N_221–235_	LLLLDRLNQLESKMS	[43]
HLA-A*02:01	N_221–230_	LLLLDRLNQL	[83]
HLA-DR	N_311–325_	ASAFFGMSRIGMEVT	[43]
NA	N_326–340_	PSGTWLTYTGAIKLD	[42]
NA	N_328–342_	GTWLTYTGAIKLDDK	[43]

Instances where cross-reactive T cell immunity from HcoVs result in cross-protective effects in SARS-CoV-2 infection are now clearly established in the literature. Further research into the relative contribution of cross-protective versus de novo immunity in combatting COVID-19 would assist in unravelling the often-convoluted history of T cell memory mixed with the somewhat plastic nature of T cell cross-reactivity. In addition, further research is required to address the interplay between cross-reactive T cell immunity and other immune cells to mount an orchestrated immune response against SARS-CoV-2.

## 5. T Cell Cross-Reactivity between SARS-CoV-2 and Novel SARS-CoV-2 Variants

As the COVID-19 pandemic progressed, novel variants began to emerge that had the capacity to increase transmission or escape pre-existing immunity to prior SARS-CoV-2 infection or vaccination. These variants included Alpha, Beta, Gamma, Delta, Mu and Omicron, with Omicron having the highest number of mutations [84]. Some of these variants were in the receptor-binding domain of the spike protein, which is a key target for neutralizing antibodies and a target for SARS-CoV-2 vaccines. These variants were less able to be controlled by neutralizing antibodies, particularly the Omicron variant [85]. Despite the concerning decrease in humoral immunity to the novel variants, memory T cells remained largely unaffected. This is partly due to the majority of T cell epitopes in the variants remaining unchanged [66,86,87,88]. Some particular epitopes, in the context of certain HLA alleles, reported a decrease in memory T cell recognition by SARS-CoV-2 variants [89,90,91,92]. This highlights that in some populations, the cross-reactive T cell repertoire from previous SARS-CoV-2 exposure or vaccination may be less able to mount effective immune responses against novel variants. However, given the already characterised breath of memory T cell repertoire for SARS-CoV-2, there is less risk of immune escape [86,93]. The low risk of immune escape was corroborated, since, in the general population, approximately 80–100% T cell cross-reactivity between the original Wuhan strain of SARS-CoV-2 and later variants was observed [66,87,94,95,96,97,98]. The influence of cross-reactive T cells on SARS-CoV-2 variants contributed to protection from severe COVID-19 after re-infection, which remained high at over 88% protection against severe disease up to 40 weeks after the first infection regardless of the variant responsible for reinfection [99]. The preservation of cross-reactive memory T cell responses to SARS-CoV-2 variants of concern has ensured that prior SARS-CoV-2 exposure or vaccination can still have clinically protective effects upon exposure to novel SARS-CoV-2 variants.

## 6. T Cell Cross-Reactivity between SARS-CoV-2 and Different Vaccines or Pathogens

Given the well-characterised involvement of cross-reactive T cells between HcoVs and SARS-CoV-2 and its variants, other sources of cross-reactivity began to emerge as potentially responsible for cross-reactive T cell immunity to SARS-CoV-2. It was found that HcoVs could not completely explain the cross-reactive memory T cell responses in unexposed individuals to SARS-CoV-2, and, therefore, T cell memory responses from other previous infections or vaccinations also contribute to the cross-reactive T cell response to SARS-CoV-2 [33,42,55,100]. Several notable contributions of memory T cell cross-reactivity between SARS-CoV-2 and the BCG vaccine, influenza A, Measles, Mumps, Rubella vaccine, *Paramyxovirus* and bacterial pathogens will be explored.

## 7. T Cell Cross-Reactivity from the Bacillus Calmette–Guérin Vaccine

Early in the COVID-19 pandemic, before SARS-CoV-2-specific vaccines were available, the heterologous BCG vaccination was used as a way to protect people from COVID-19, especially high-risk groups such as frontline healthcare workers and the elderly [101,102]. The heterologous effects of the BCG vaccination have been widely studied, which involved heterologous CD4+ T cell immunity and trained innate immunity, leading to a reduction in all-cause mortality in BCG-vaccinated children and a reduction in respiratory tract infections in adults [103,104,105,106]. Mouse studies have shown that BCG can protect against SARS-CoV-2 and influenza infection via the engagement of the innate and adaptive immune system, particularly CD4+ T helper cells [107]. Clinical trials assessing the outcome of SARS-CoV-2 infection in BCG-vaccinated individuals showed mixed results (Table 2) with 10 trials and retrospective observational studies showing a protective effect (NCT04659941, NCT04369794, NCT04414267, NCT04417335, CTRI/2020/05/025013, NCT04475302, CTRI/2020/07/026668) [108,109,110,111,112,113,114,115,116,117], whereas 7 trials showed no protective effect (NCT04373291, RBR-4kjqtg, NCT04328441, NCT04537663, NCT04648800, NCT04379336, NCT04327206) [118,119,120,121,122,123]. Each study looked at the protective effect that the BCG vaccination has for COVID-19 in different ways, and each study assessed different populations, which may explain the mixed results between trials. Overall, the clinical trials showed that BCG vaccination prior to SARS-CoV-2 infection can induce heterologous immunity including heterologous T cell and antibody responses, which, in some instances, improved COVID-19 outcomes. The development of SARS-CoV-2-specific vaccinations and their global administration and high efficacy has led to a shift away from using the BCG vaccination for protecting against COVID-19.

Increasingly, the involvement of cross-reactive memory T cells is becoming understood as influencing the effect of the BCG vaccination on SARS-CoV-2 infection or vaccination. The analysis of epitopes from BCG proteins has uncovered significant homology to many SARS-CoV-2 epitopes [124,125,126]. There is evidence to suggest that BCG-specific memory T cells can cross-react with SARS-CoV-2-presented epitopes in a TCR-dependent manner [125]. A clinical trial where young adult participants received BCG re-vaccination, followed by SARS-CoV-2 vaccination, showed evidence of an increased benefit from receiving the BCG re-vaccination through enhanced immune responses [127]. In this study, a hallmark of antigen-specific, TCR-dependent memory T cell responses by activation-induced markers (AIM) was observed to be increased in CD4+ and CD8+ memory T cells and BCG-re-vaccinated and SARS-CoV-2-vaccinated individuals. This suggests that TCR-dependent activation of BCG-specific memory T cells by SARS-CoV-2 vaccination may be responsible for enhancing immune responses to SARS-CoV-2 vaccination. The clinical relevance of the TCR-dependent cross-reactivity between the BCG vaccine and COVID-19 has not been fully explored in the completed clinical trials or retrospective observational studies, and as such, further research is required to understand whether this phenomenon is a correlate of protection.

Cross-reactive T cells have been shown to be implicated in reducing the severity of COVID-19 outcomes. In blood samples from a clinical trial, those that received the BCG vaccination and then stimulated with SARS-CoV-2 produced fewer hallmarks of severe COVID-19 through cytokine profiling compared to placebo-vaccination [128]. Additionally, from the same study, BCG vaccination and SARS-CoV-2 stimulation increased the proportion of CD4+ T effector memory cells and CD8+ TEMRA cells and decreased the proportion of naïve T cells compared to placebo vaccination [128]. Another clinical trial where participants received the BCG vaccination then SARS-CoV-2 vaccination showed evidence of an increased benefit from receiving the BCG vaccination [127]. Poly-functional, cross-reactive memory T cells were significantly higher in participants who received the BCG vaccination before the SARS-CoV-2 vaccination, with CD4+ T effector memory and CD8+ TEMRA again being involved. Similar enhanced frequencies of memory T cells were observed in BCG-vaccinated elderly individuals, suggesting that the BCG vaccine can also induce poly-specific memory T cell responses in elderly patients who are at heightened risk of experiencing severe COVID-19 [129]. Overall, this suggests that the memory T cells produced from BCG-vaccination cross-react upon exposure to SARS-CoV-2 infection or vaccination.

## 8. T Cell Cross-Reactivity from the Influenza Vaccine

Early on in the pandemic, an association between high influenza vaccination coverage and lower SARS-CoV-2 infection rates was observed [130,131,132,133]. Further research showed that influenza A virus epitopes could generate memory T cells that cross-react with SARS-CoV-2 epitopes [35]. This was shown through shared TCR clonotypes and cross-reactive functional cytokine responses between SARS-CoV-2 and influenza A virus epitopes. Therefore, vaccination or exposure to influenza A virus may generate cross-reactive memory T cells that can influence the immune response to SARS-CoV-2 or vice versa. Further research is required to discover whether such cross-reactive T cells are a correlate of protection in COVID-19.

## 9. T Cell Cross-Reactivity from the Measles, Mumps and Rubella Vaccine

Studies have shown that the measles, mumps and rubella (MMR) vaccine is associated with a better COVID-19 outcome [36,134,135]. Insights into the mechanism behind such cross-protection revealed that individuals vaccinated with the MMR or tetanus, diphtheria and pertussis (Tdap) vaccine shared TCR clonotypes with individuals who were convalescent or vaccinated against SARS-CoV-2 [36]. Furthermore, sequence homologies between MMR surface proteins and SARS-CoV-2 spike have been identified that may give rise to cross-reactive T cells [136,137]. This suggests that T cell cross-reactivity between MMR or Tdap and SARS-CoV-2 may result in a cross-protective effect against COVID-19. More research into the magnitude of the protective role of MMR and Tdap vaccines in terms of SARS-CoV-2 is required.

## 10. T Cell Cross-Reactivity from Microbial Antigens

Microbial antigens, both pathogenic and commensal, have been shown to exhibit homologies with known epitopes of SARS-CoV-2, giving rise for the potential for cross-reactive memory T cells to be involved in the SARS-CoV-2 immune response [138,139]. This shared homology has previously been shown to generate cross-reactive memory T cells that initially arose from exposure to a bacterial antigen and could become activated after exposure to SARS-CoV-2 epitopes [138,139]. In one study, these memory T cells expressed gut-homing markers, highlighting that they likely arise from microbial antigens from common commensal bacteria [139]. Thus, bacteria may be a source of memory T cells that can cross-react upon exposure to SARS-CoV-2.

Overall, the cross-reactivity from pathogens unrelated to HcoVs may add to the overall cross-reactive memory T cell response in SARS-CoV-2 infection. Further research is needed to understand the contribution of the identified cross-reactive immune responses to overall clinical outcomes in COVID-19 patients.

## 11. Conclusions and Future Directions

In this study, we have provided an up-to-date account of the mechanisms and role of T cell cross-reactivity in SARS-CoV-2 infection. Such cross-reactivity can arise from pre-exposure to a variety of heterologous pathogens or vaccines. The case for cross-reactive T cell immunity between seasonal HCoVs and SARS-CoV-2 is well established with some degree of cross-protective benefit. A mechanism for T cell cross-reactivity between heterologous vaccinations or other pathogens and SARS-CoV-2 has been established. Further research is required to determine whether the identified cross-reactive T cells from heterologous vaccinations or other pathogens are a correlate of protection in COVID-19.

## Figures and Tables

**Figure 1 biomedicines-12-00564-f001:**
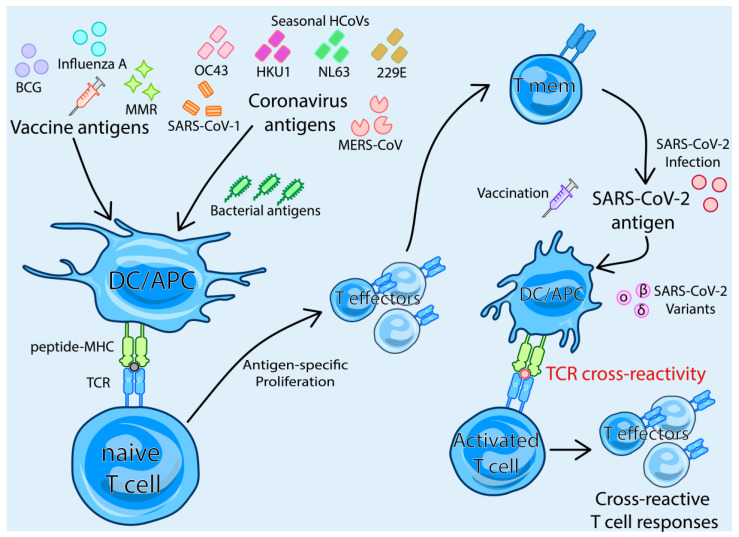
Mechanism of TCR-dependent T cell cross-reactivity with SARS-CoV-2.

**Table 2 biomedicines-12-00564-t002:** Clinical trials of the heterologous BCG vaccination for COVID-19.

Registry Number	Study Title	Phase/Country/Participant Group	Outcome
NCT04659941	Use of BCG Vaccine as a Preventive Measure for COVID-19 in Health Care Workers (ProBCG)	Phase 2 Brazil Healthcare workers	BCG could protect from COVID-19 [109]
RBR-4kjqtg	BCG revaccination of health care professionals working in the COVID-19 pandemic, a preventive strategy to improve innate immune response	Phase 2 Brazil Healthcare workers	BCG could not protect from COVID-19 [119]
NCT04373291	Using BCG Vaccine to Protect Health Care Workers in the COVID-19 Pandemic	Phase 3 Denmark Healthcare workers	BCG could not protect from COVID-19 [118]
NCT04414267	Bacillus Calmette-guérin Vaccination to Prevent COVID-19 (ACTIVATEII)	Phase 4 Greece Adults ≥ 50 years with comorbidities	BCG could protect from COVID-19 [110]
NCT04328441	Reducing Health Care Workers Absenteeism in COVID-19 Pandemic Through BCG Vaccine (BCG-CORONA)	Phase 3 Netherlands Healthcare workers	BCG could not protect from COVID-19 [120]
NCT04417335	Reducing COVID-19 Related Hospital Admission in Elderly by BCG Vaccination	Phase 4 Netherlands Adults ≥ 60 years	BCG could protect from COVID-19. [111]
NCT04537663	Prevention Of Respiratory Tract Infection And COVID-19 Through BCG Vaccination In Vulnerable Older Adults (BCG-PRIME)	Phase 4 Netherlands Adults ≥ 60 years with comorbidities	BCG could not protect
NCT04648800	Clinical Trial Evaluating the Effect of BCG Vaccination on the Incidence and Severity of SARS-CoV-2 Infections Among Healthcare Professionals During the COVID-19 Pandemic in Poland	Phase 3 Poland Healthcare workers	BCG could not protect from COVID-19 [121]
CTRI/2020/05/025013	Phase 2 Clinical Trial for the Evaluation of BCG as potential therapy for COVID-I9	Phase 2 India Adults with COVID-19	BCG could protect from COVID-19 [112]
NCT04475302	BCG Vaccine in Reducing Morbidity and Mortality in Elderly Individuals in COVID-19 Hotspots	Phase 3 India Adults 60–80 years	BCG could protect from COVID-19 [113]
CTRI/2020/07/026668	To study the effect of BCG vaccine in Reducing the Incidence and severity of COVID-19 in the high-risk population	Phase N/A India High-risk groups of adults 18–60 years	BCG could protect from COVID-19 [114]
NCT04379336	BCG Vaccination for Healthcare Workers in COVID-19 Pandemic	Phase 3 South Africa Healthcare workers	BCG could not protect from COVID-19 [122]
NCT04327206	BCG Vaccination to Protect Healthcare Workers Against COVID-19 (BRACE)	Phase 3 Australia and Netherlands Healthcare workers	BCG could not protect from COVID-19 [123]
NCT04369794	COVID-19: BCG As Therapeutic Vaccine, Transmission Limitation, and Immunoglobulin Enhancement (BATTLE)	Phase 4 Brazil	BCG could protect from COVID-19 [108]

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
