# Peer review of "The Influence of Cross-Reactive T Cells in COVID-19"

_biomedicines, 2024, doi:10.3390/biomedicines12030564_

Round 1
Reviewer 1 Report
Comments and Suggestions for Authors
The manuscript reviews available evidence about T-cell cross-reactivity between SARS-CoV-2 and other coronaviruses in the context of COVID-19, its infection risk and the risk of severe disease. The manuscript is logicaly organised, comprehensive and well writen.
Author Response
We thank the reviewer for their thoughtful appraisal of the manuscript.
Reviewer 2 Report
Comments and Suggestions for Authors
The manuscript provides an overview on activation of cross-reactive and bystander T cells during COVID-19 infection.
The review is comprehensive and well written.
I would suggest to use a more appropriate terminology to described the 2 types of T cell activation.
I would not describe the TCR independent T cell activation (bystander) as T cell cross-reactivity and use the word cross reactivity to describe only the TCR depended phenomenon. Moreover, since the bystander activation is not in the scope of this review I would not have a dedicated section but may be just describe this phenomenon in the introduction.
Author Response
We agree with the reviewer and have removed reference to ‘TCR-independent cross-reactivity. We removed section 3 on ‘TCR-Independent T cell Cross-Reactivity’. As suggested, we have now described bystander effects in section 2, ‘Adaptive Immune Response to SARS-CoV-2’.
Reviewer 3 Report
Comments and Suggestions for Authors
Eggenhuizen et al's review "The Influence of Cross-Reactive T cells in COVID-19" is a timely and well-written review summarizing the updates of T cell-mediated cross-reactivity against COVID-19, despite some parts being of potential to be further improved.
1 The definition of "TCR-Independent T cell Cross-Reactivity" is not appropriate, in my opinion. Usually, cross-reactivity should be TCR-dependent since TCR is the receptor used by T cells to specifically recognize antigen. The classical "by-stander effects" is better in describing the phenomenon discussed in this section.
2 A table listing potential epitopes or peptides recognized by cross-reactive T cells identified in COVID-19 (human or mouse) and their corresponding MHC types will be very helpful and thus be recommended strongly.
3 Fig.1 shows only the general T cell responses. The authors may want to make it more "cross-reactive" by adding mechanisms described in section 5 "T cell Cross-Reactivity between SARS-CoV-2 and Other Human Coronaviruses".
4 Text of section 4 "TCR-Dependent Cross-Reactivity" is redundant (eg. Lines 90-94 is quite similar to Lines 104-107) and can be refined into a single paragraph.
Author Response
Eggenhuizen et al's review "The Influence of Cross-Reactive T cells in COVID-19" is a timely and well-written review summarizing the updates of T cell-mediated cross-reactivity against COVID-19, despite some parts being of potential to be further improved.
1 The definition of "TCR-Independent T cell Cross-Reactivity" is not appropriate, in my opinion. Usually, cross-reactivity should be TCR-dependent since TCR is the receptor used by T cells to specifically recognize antigen. The classical "by-stander effects" is better in describing the phenomenon discussed in this section.
We agree with the reviewer and to maintain the focus on the main topic we have removed section 3 on TCR-independent T cell cross-reactivity. We have removed reference to TCR-independent T cell cross reactivity and have instead covered bystander effects in section 2, ‘Adaptive Immune Response to SARS-CoV-2’.
2 A table listing potential epitopes or peptides recognized by cross-reactive T cells identified in COVID-19 (human or mouse) and their corresponding MHC types will be very helpful and thus be recommended strongly.
We thank the reviewer for their constructive comment and have devised a Table illustrating the SARS-CoV-2 T cell epitopes that have been shown to cross-react with human coronaviruses. Where the HLA-type was identified in the paper, we have included this in the table as well. This is now new ‘Table 1’.
3 Fig.1 shows only the general T cell responses. The authors may want to make it more "cross-reactive" by adding mechanisms described in section 5 "T cell Cross-Reactivity between SARS-CoV-2 and Other Human Coronaviruses".
We thank the reviewer for the suggestion to make the figure more ‘cross-reactive’ to highlight the mechanisms between SARS-CoV-2 and other human coronaviruses. To address this, we have added extra detail in the figure around the human coronaviruses, see new updated figure.
4 Text of section 4 "TCR-Dependent Cross-Reactivity" is redundant (eg. Lines 90-94 is quite similar to Lines 104-107) and can be refined into a single paragraph.
Lines 104-107 were removed as the reviewer highlighted it was already covered in Lines 90-94.